# Testing strategies to detect acute and prevalent HIV infection in adult outpatients seeking healthcare for symptoms compatible with acute HIV infection in Kenya: a cost-effectiveness analysis

Joseph B Babigumira ![ORCID],[1] Clara A Agutu,[2] Deven T Hamilton,[3] Elise van der Elst,[2] Amin Hassan,[2] Evanson Gichuru,[2] Peter Mwangi Mugo ![ORCID],[2] Carey Farquhar,[4] Thumbi Ndung'u,[5] Martin Sirengo,[6] Wairimu Chege,[7] Steven M Goodreau,[8] Eduard J Sanders,[2,9] Susan M Graham ![ORCID] [2,10]

For numbered affiliations see end of article.

**Correspondence to**
Dr Joseph B Babigumira;
josephbabigumira@gmail.com

## ABSTRACT

**Background** Detection of acute and prevalent HIV infection using point-of-care nucleic acid amplification testing (POC-NAAT) among outpatients with symptoms compatible with acute HIV is critical to HIV prevention, but it is not clear if it is cost-effective compared with existing HIV testing strategies.

**Methods** We developed and parametrised a decision tree to compare the cost-effectiveness of (1) provider-initiated testing and counselling (PITC) using rapid tests, the standard of care; (2) scaled-up provider-initiated testing and counselling (SU-PITC) in which all patients were tested with rapid tests unless they opted out; and (3) opt-out testing and counselling using POC-NAAT, which detects both acute and prevalent infection. The model-based analysis used data from the Tambua Mapema Plus randomised controlled trial of a POC-NAAT intervention in Kenya, supplemented with results from a stochastic, agent-based network model of HIV-1 transmission and data from published literature. The analysis was conducted from the perspective of the Kenyan government using a primary outcome of cost per disability-adjusted life-year (DALY) averted over a 10-year time horizon.

**Results** After analysing the decision-analytical model, the average per patient cost of POC-NAAT was $214.9 compared with $173.6 for SU-PITC and $47.3 for PITC. The mean DALYs accumulated per patient for POC-NAAT were 0.160 compared with 0.176 for SU-PITC and 0.214 for PITC. In the incremental analysis, SU-PITC was eliminated due to extended dominance, and the incremental cost-effectiveness ratio (ICER) comparing POC-NAAT to PITC was $3098 per DALY averted. The ICER was sensitive to disability weights for HIV/AIDS and the costs of antiretroviral therapy.

**Conclusion** POC-NAAT offered to adult outpatients in Kenya who present for care with symptoms compatible with AHI is cost-effective and should be considered for inclusion as the standard of HIV testing in this population.

## STRENGTHS AND LIMITATIONS OF THIS STUDY

⇒ To our knowledge, this is the first study in sub-Saharan Africa to consider the cost-effectiveness of nucleic acid amplification testing for acute HIV infection (AHI) in symptomatic outpatients.

⇒ The study included three possible alternatives for acute HIV testing in symptomatic outpatients to maximise local policy relevance.

⇒ The study did not include alternative methods for the diagnosis of AHI such as rapid antigen–antibody tests.

⇒ The study time horizon was limited to 10 years instead of a lifetime, favouring short-term certainty over long-term uncertainty.

**Trial registration number** Tambua Mapema ("Discover Early") Plus study (NCT03508908) conducted in Kenya (2017–2020) i.e., Post-results.

## INTRODUCTION

HIV testing is the gateway to lifelong treatment of HIV-positive individuals with antiretroviral therapy (ART), which prolongs life, reduces morbidity and optimises quality of life. HIV testing is also the gateway to HIV prevention through knowledge of HIV status, which empowers HIV-negative individuals to protect themselves, and to treatment as prevention, which effectively renders treated individuals who achieve virological suppression unable to transmit to others.[1] In 2015, Joint United Nations Programme on HIV/AIDS (UNAIDS) set a target of achieving a 90% rate of diagnosis of people living with HIV (PLWH) by 2020 and a 95% rate of

diagnosis of PLWH by 2030, and put both metrics on a 'fast track' to ending the AIDS epidemic as a public health threat.[2] However, global progress has not achieved the desired goals: only 81% of PLWH knew their status in 2019.[3]

Although Kenya achieved the 90% diagnosis goal for PLWH by the end of 2020,[3] reaching the 95% target will be challenging because the current HIV rapid antibody detection tests in use in Kenya are unable to diagnose acute HIV infection (AHI), present during the period before seroconversion.[4] In Kenya, most patients with AHI develop a febrile illness and present to health facilities for urgent care.[5] This provides an opportunity for early diagnosis of AHI at outpatient clinics using newly available nucleic acid amplification testing (NAAT) technologies.[6] Diagnosis of AHI with linkage to care and prompt treatment with ART would lead to important downstream treatment and prevention benefits. Testing for AHI has significant HIV prevention implications because the AHI period is a period of high risk for secondary HIV transmission due to very high viral load.[7] While there are advantages of NAAT in outpatient health facilities in resource-limited settings, these tests remain costly, sometimes up to 20 times the cost of rapid tests on a per-test basis. Consequently, there has been a lack of effectiveness and cost-effectiveness evidence to support the widespread use of NAAT testing in the outpatient setting in Kenya and other sub-Saharan African countries.

To address the effectiveness gap in knowledge, the Tambua Mapema Plus (TMP)[8] study, a modified stepped wedge randomised controlled trial, was conducted. The TMP trial assessed the yield of an opt-out point-of-care nucleic acid amplification testing (POC-NAAT) intervention using Cepheid's Xpert HIV-1 Qual to diagnose acute and chronic HIV infections in Kenya.[8] In the trial, use of opt-out POC-NAAT resulted in a twofold greater odds of HIV diagnosis compared with standard of care, that is, provider-initiated HIV-1 antibody testing.[8] The TMP trial also provided the parametrisation basis for the development of a stochastic, agent-based network model to simulate the HIV epidemic in Kenya and quantify the population-level prevention benefits of POC-NAAT testing.[9] The modelling study included a scaled-up provider-initiated testing strategy in which all eligible outpatients were modelled to receive rapid HIV tests unless they opted out. The model projected that using opt-out POC-NAAT was superior to provider-initiated testing or scaled-up HIV rapid testing in terms of both yield of HIV positives (knowledge of status) and HIV prevention.[9] In this study, we present the results of a cost-effectiveness analysis to address the gap in economic evidence to further support the widespread use of NAAT testing for AHI in the outpatient setting in Kenya and other sub-Saharan African countries.

## METHODS

It was not appropriate or possible to involve patients or the public in the design, conduct, reporting or dissemination plans of our research.

### Study design

We conducted a synthesis-based and model-based cost-effectiveness analysis using costs and outcomes data obtained from the TMP trial,[10 11] supplemented by data from a stochastic, agent-based mathematical modelling study[9] and additional data from the published literature. The mathematical model was developed to evaluate the population-level and HIV prevention impact of the TMP intervention and was parametrised using data from the TMP trial.[9] Data from the published literature were used to supplement cost and outcome data obtained from the trial and outcome data obtained from the mathematical model (table 1).

### Target population

The base case population in the analysis included adults 18–39 years, not previously diagnosed with HIV, who presented as outpatients to six public and private primary care health facilities in Kenya with one or more symptoms of AHI. Patients qualified for inclusion in the trial, and consequently in this study, if they had a score of ≥2 on a risk screening score defined by younger age (18–29 years), fever, fatigue, body pains, diarrhoea and sore throat (one point each), or genital ulcer disease (three points).[10 12]

### Setting

The study setting and decision-making context was peri-urban coastal Kenya (Mombasa and Kilifi counties), an area with busy nightlife, sex work and tourism (estimated HIV prevalence of 5.6% and 2.3% for Mombasa and Kilifi counties, respectively, in 2018[13]). Therefore, insights from this study may also have decision-making relevance in urban and periurban settings in Kenya and other, similar low-income, high-HIV burden countries.

### Comparators

We compared three HIV testing strategies as follows:
1. Provider-initiated testing and counselling (PITC), the standard of current care in Kenya. Under PITC, providers ordered rapid HIV-1 antibody testing at their discretion as a diagnostic aid. This comparator is identical to the observation arm in the TMP trial, in which only 352 (25.6%) of 1374 enrolled participants were tested for HIV and 13 (0.9%) of 1374 enrolled participants were newly diagnosed with HIV.[14]
2. Scaled-up provider-initiated testing and counselling (SU-PITC), in which providers were modelled to order rapid HIV-1 antibody testing for all patients unless they opted out of HIV testing. This comparator was included as a potentially plausible policy alternative for testing among this group of outpatients, given that it excludes relatively costly POC-NAAT.
3. POC-NAAT testing and counselling using point-of-care Xpert HIV-1 Qual, as performed under the

**Table 1** Model inputs: probabilities, outcomes estimates, resource estimates and cost parameters

|  | Base case | Low | High | Reference |
|---|---|---|---|---|
| **Probabilities** |  |  |  |  |
| Provider-recommended HIV Ab testing | 0.256 | 0.205 | 0.307 | Primary data[9] |
| Opt-out |  |  |  |  |
| Of physician-recommended Ab test (PITC) | 0.000 | — | — | Primary data[9] |
| Of scaled-up Ab test (SU-PITC) | 0.051 | 0.041 | 0.061 | Primary data*[9] |
| Of RNA test (POC-NAAT) | 0.051 | 0.041 | 0.061 | Primary data[9] |
| HIV prevalence in Kenya | 0.049 | 0.041 | 0.210 | [13] |
| Linkage to care |  |  |  |  |
| PITC | 0.692 | 0.554 | 0.830 | Primary data[9] |
| POC-NAAT/SU-PITC | 0.892 | 0.714 | 1.000 | Primary data*[9] |
| Partner disclosure |  |  |  |  |
| PITC | 0.75 | 0.600 | 0.900 | Primary data[9] |
| POC-NAAT /SU-PITC | 0.67 | 0.536 | 0.804 | Primary data*[9] |
| Partner positivity |  |  |  |  |
| If true positive | 0.230 | 0.184 | 0.276 | [38] |
| If false positive | 0.049 | 0.041 | 0.210 | †[13] |
| Disability weights |  |  |  |  |
| HIV (symptomatic, pre-AIDS) | 0.274 | 0.184 | 0.377 | [21] |
| HIV/AIDS (receiving ART) | 0.078 | 0.052 | 0.111 | [21] |
| **Cost inputs** |  |  |  |  |
| GeneXpert |  |  |  |  |
| Electricity consumption (VA) | 824 | — | — | Cepheid |
| Cost per unit ($) (KWA) of electricity | 0.14 | 0.07 | 0.29 | [39] |
| Kenya voltage | 240 | — | — | Standard |
| Power factor | 0.6 | 0.48 | 0.72 | Assumption |
| Machine ($) (GeneXpert IV) | 17 000 | 13 600 | 20 400 | [40] |
| Warranty ($) | 6480 | 5184 | 7776 | [40] |
| Cartridge ($) (per test) | 16.80 | 13.44 | 20.16 | [40] |
| Maintenance ($) (per year) | 1800 | 1440 | 2160 | [40] |
| Useful life (years) | 5 | 4 | 6 | [41] |
| Scrap value ($) | 0 | – | – | Assumption |
| Tests per hour | 2.5 | 2.0 | 3.0 | [40] |
| Annual run time (hours) | 2848 | 2278 | 3418 | Assumption |
| Determine ($) | 0.82 | 0.41 | 1.64 | [42] |
| First response ($) | 0.73 | 0.36 | 1.46 | [42] |
| Personnel (cadres) by activity |  |  |  |  |
| Clinical consultation (%) |  |  |  |  |
| Medical officers | 12.5 | 8 | 12 | Primary data |
| Clinical officers | 87.5 | 72% | 100 | Primary data |
| Lab testing (%) |  |  |  |  |
| Lab technologist | 95 | 76 | 100 | Primary data |
| Medical lab officer | 5 | 4 | 6 | Primary data |
| Pharmacy services (%) |  |  |  |  |
| Clinical officer | 5 | 4 | 6 | Primary data |
| Pharmacist | 10 | 8 | 12 | Primary data |
| Pharmacy technologist | 85 | 68 | 100 | Primary data |
| Time use (min) |  |  |  |  |

**Table 1** Continued

| | Base case | Low | High | Reference |
|---|---|---|---|---|
| Observation arm | | | | |
| Initial consultation | 4 | 3 | 5 | Primary data |
| Lab testing | 54 | 31 | 85 | Primary data |
| Postlab consultation | 4 | 1 | 8 | Primary data |
| Pharmacy | 2 | 1 | 5 | Primary data |
| Intervention arm | | | | |
| Initial consultation | 3 | 3 | 5 | Primary data |
| Pretest counselling | 17 | 14 | 22 | Primary data |
| Lab testing | 68 | 45 | 93 | Primary data |
| GeneXpert testing | 110 | 101 | 144 | Primary data |
| Post-test counselling | 3 | 2 | 4 | Primary data |
| Postlab consultation | 8 | 5 | 14 | Primary data |
| Pharmacy | 5 | 3 | 7 | Primary data |
| Wages ($) (per hour) | | | | |
| Medical officers | 1.04 | 0.39 | 1.56 | Primary data‡ |
| Clinical officers | 0.50 | 0.33 | 0.75 | Primary data‡ |
| Lab technologist | 0.36 | 0.18 | 0.54 | Primary data‡ |
| Medical lab officer | 0.47 | 0.36 | 0.71 | Primary data‡ |
| Pharmacist | 1.08 | 0.39 | 1.62 | Primary data‡ |
| Pharmacy technologist | 0.35 | 0.33 | 0.52 | Primary data‡ |
| Counsellor | 0.47 | 0.33 | 0.71 | Primary data‡ |
| Linkage to HIV care ($)§ | 225.45 | 112.73 | 338.18 | [17] |
| Antiretroviral therapy ($) | | | | |
| Initiation | 102.54 | 51.27 | 153.81 | [17] |
| Maintenance (annual) | 351.10 | 175.55 | 526.65 | [17] |
| Discounted LYs (10-year time horizon) | 8.53 | | | Calculated |
| PrEP (annual) (%) | 111.29 | 83.47 | 166.94 | [18] |
| Duration of PrEP (years) | 0.315 | 0.252 | 0.378 | Kenya MOH |
| Acute partner services (index positive) ($) | 114.39 | 57.20 | 171.59 | [17] |

*Assumption: opt out rate, rate of linkage to care and rate of partner disclosure in the SU-PITC arm are equivalent to rates in the POC-NAAT arm.
†Assumption: equal to HIV prevalence.
‡Lower estimate equals starting wage for cadre.
§Cost of pre-ART laboratory tests assuming all patients have a CD4 cell count of over 350 cells/µL.
ART, antiretroviral therap; KWA, kilowatt hour; LY, life-year; PITC, provider-initiated testing and counselling; POC-NAAT, point-of-care nucleic acid amplification testing; PrEP, pre-exposure prophylaxis; SU-PITC, scaled-up provider-initiated testing and counselling.

intervention arm of the TMP trial. Under this POC-NAAT approach, providers ordered NAAT testing for all included patients except those who were eligible but opted out of the study. Patients who tested positive by POC-NAAT received rapid HIV-1 antibody testing to distinguish between acute and prevalent HIV infection. This comparator is identical to the intervention arm in the TMP trial, in which all 1500 participants were tested for HIV, and 37 (2.5%) were newly diagnosed with HIV, of whom two were AHI cases.[11]

## Time horizon
The analysis was conducted over a time horizon of 10 years, consistent with the mathematical model,[9] and thereby included the 10-year costs of ART for individuals

who tested positive for HIV and costs of pre-exposure prophylaxis (PrEP) for HIV-negative partners of newly diagnosed index patients that were based on the mean duration of PrEP use in Kenya (table 1).

## Decision analytical model
We developed a decision-analytical model (decision tree) to assess the cost-effectiveness of the three different HIV testing strategies. Figures 1–3 show the decision tree branches proceeding from the decision node representing the choice between PITC (figure 1), SU-PITC (figure 2) and POC-NAAT (figure 3). The data used to parametrise the model, including ranges used for sensitivity analysis, are shown in detail in table 1. In PITC, base case patients received a determine HIV antibody test

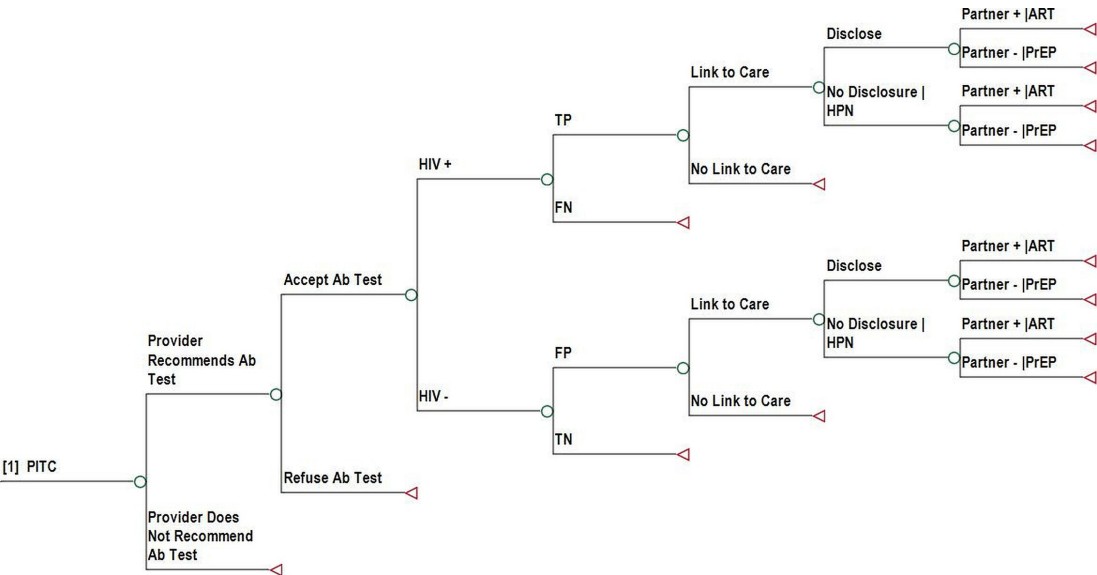

**Figure 1** Decision tree branch showing the consequences of PITC. The decision tree branch follows a decision node. Ab, antibody; ART, antiretroviral therapy; FN, false negative; FP, false positive; HPN, HIV partner notification; PITC, provider-initiated testing and counselling; PrEP, pre-exposure prophylaxis; TN, true negative; TP, true positive.

at the recommendation and discretion of the provider unless they refused HIV testing. In SU-PITC, all patients received a Determine HIV antibody test unless they opted out. Patients who tested positive with Determine in either scenario received confirmatory testing with the first response HIV antibody test. In POC-NAAT, patients who had positive HIV-1 RNA Qual results received HIV rapid testing with the two different antibody tests in parallel, to distinguish acute from prevalent HIV infection. In each study arm, HIV rapid tests were repeated for confirmation of diagnosis prior to ART initiation, in accordance with Kenyan guidelines.[15]

The decision tree then divided patients into HIV-positive and HIV-negative based on the prevalence of HIV in Kenya.[13] The decision tree assessed the accuracy of testing strategies by dividing patient test results into true positive (TP), false negative (FN), false positive (FP) and true negative (TN), based on estimates from the mathematical model.[9] The decision tree simulated the experience of base case patients and their likelihood of ending up in the four diagnostic accuracy groups, given modelled HIV transmission risk, rates of HIV rapid testing in the outpatient setting and background rates of HIV testing in all settings over a period of 10 years. Patients with FN samples exited the decision tree under the assumption that they would receive a delayed diagnosis of HIV outside of the TMP setting (ie, outside the experience of base case patients). Patients with TN samples exited the decision tree.

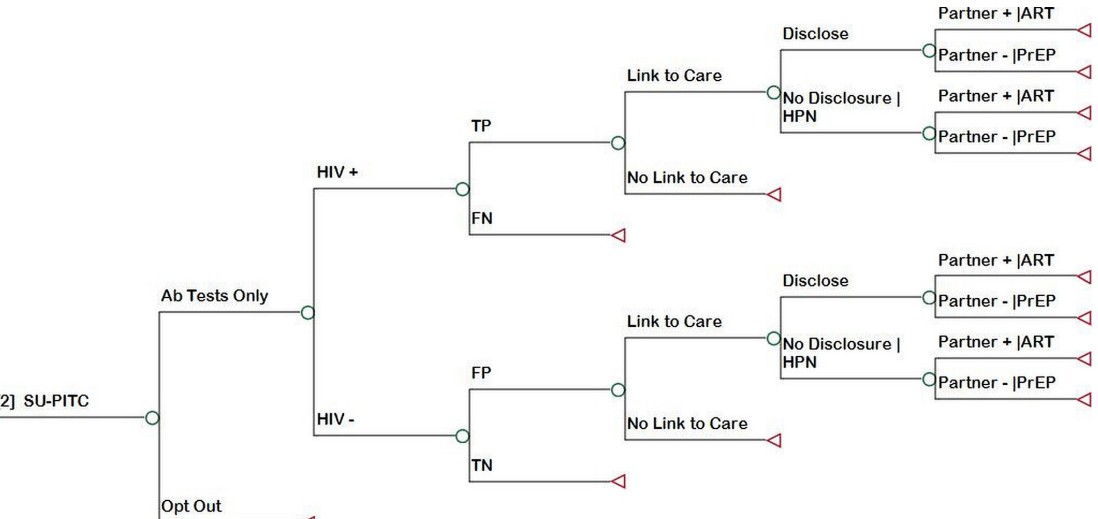

**Figure 2** Decision tree branch showing the consequences of SU-PITC. The decision tree branch follows a decision node. Ab, antibody; ART, antiretroviral therapy; FN, false negative; FP, false positive;; HPN, HIV partner notification; PrEP, pre-exposure prophylaxis; SU-PITC, scaled-up provider-initiated testing and counselling; TN, true negative; TP, true positive.

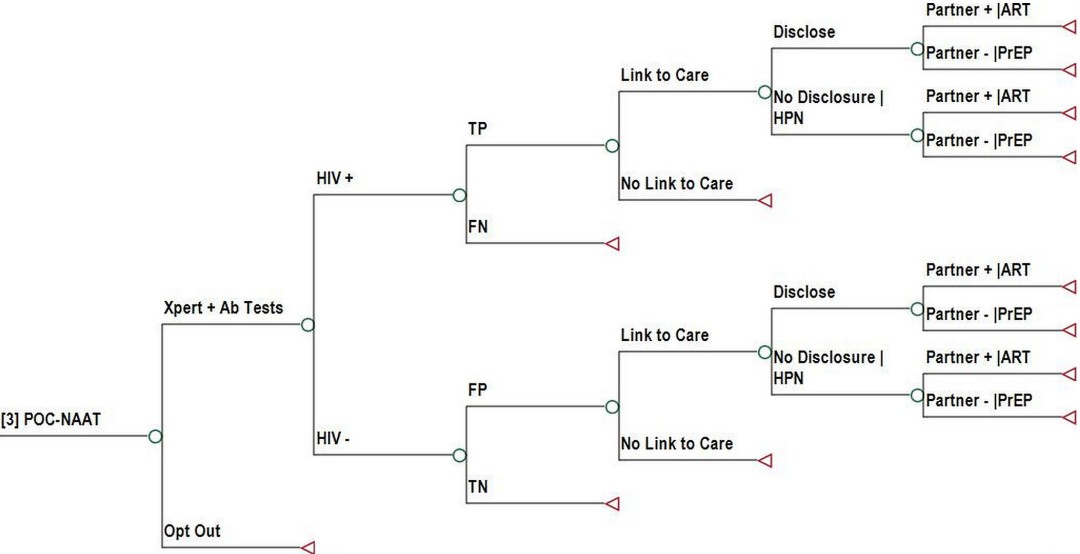

**Figure 3** Decision tree branch showing the consequences of POC-NAAT. The decision tree branch follows a decision node. Ab, antibody; ART, antiretroviral therapy; FN, false negative; FP, false positive; HPN, HIV partner notification; POC-NAAT, point-of-care nucleic acid amplification testing; PrEP, pre-exposure prophylaxis; TN, true negative; TP, true positive.

Patients with TP and FP test results were assumed and modelled in the decision tree to be linked to care or not and, if linked to care, received ART for the model time horizon. They were then modelled in the decision tree to disclose their HIV status to partners or not. Those who did not disclose to partners entered the decision tree branch in which they received HIV partner notification (HPN). Positive partners entered the decision tree branch in which they received ART for the model time horizon and negative partners (discordant negative) entered the decision tree branch in which they received PrEP for the mean duration of PrEP in Kenya.

## Outcomes

In addition to the estimation of the mean costs of the comparators over the time horizon of the study, we included four outcomes or effectiveness measures as follows: (1) correct diagnostic classification of patients, a factor of both the provider and patient decision to test and the accuracy of the diagnostic testing strategy; (2) HIV case yield; (3) HIV cases (and cases averted); and (4) disability-adjusted life-years (DALYs), the primary outcome in the analysis. The TMP trial was designed to assess the yield of acute and prevalent HIV cases by Xpert RNA testing, as well as linkage and partner outcomes at 6 weeks of follow-up.[10] Data from the trial, the mathematical model and the published literature were used to assess test accuracy and HIV case yield (table 1). The mathematical model allowed for the simulation of the potential prevention impact of the TMP intervention at scale over a 10-year time horizon.[9]

## Perspective

The perspective of the analysis was that of the government of Kenya through the Ministry of Health. Although a significant proportion of the Kenyan population pays for healthcare out of pocket, over 60% of total health expenditure is by the government, which makes it a de facto national payer.[16] As such, the governmental perspective, which includes all formal health sector (medical) costs but excludes all costs that the government would not incur at scale, is important for purposes of government policy making.

## Resource use and cost

We divided the cost estimates into two main categories: HIV-testing-related costs and costs related to the management of patients who tested positive for HIV. The costs of HIV testing included the costs of testing supplies and personnel, and the costs of HIV-positive patient management included costs of linkage to care, HPN, initiation of ART, maintenance of ART and PrEP for seronegative partners of patients newly diagnosed through the intervention. The costs of HPN, linkage to care, ART and PrEP were obtained from published studies conducted in Kenya.[17 18]

The costs of testing supplies included the costs of procurement, warranty, maintenance and electric consumption of the Cepheid GeneXpert HIV-1 POC-NAAT platform. We annuitised the capital cost of the machine and warranty over its estimated useful life and estimated the per-test cost by accounting for throughput of tests per year. We estimated the cost of GeneXpert HIV-1 qual cartridges and the Determine and First Response antibody tests on a per-test basis.

To estimate personnel costs, we conducted a primary time motion survey in which we estimated mean service times for pretest medical consultation, laboratory testing, pretest and post-test counselling (for the POC-NAAT arm), and pharmacy services (survey instrument in online supplemental material). The PITC and SU-PITC

arms of the analysis do not include the personnel costs of pretest and post-test counselling, as pretest counselling and post-test counselling were not a part of the PITC arm in the TMP trial and under the assumption that a future SU-PITC activity would proceed without dedicated pretest and post-test counselling services. Time estimates were multiplied by wages for consultants (physicians, clinical officer or nursing officer), laboratory technicians, pharmacy workers and counsellors.

### Currency, price, date and conversion

Costs were in 2019 US dollar. Cost estimates from years other than 2019 were converted to 2019 costs using the Consumer Price Index for health in Kenya.[19] Costs obtained in local currency units (Kenya shillings) were converted to US dollar using the Kenya Central Bank rate on 1 July 2019 (US$1=KSH 101.99).

### Discount rate

We used a discount rate of 3% for costs and outcomes as recommended by the second Panel on Cost-Effectiveness in Health and Medicine.[20] This discount rate applies to the annuitisation of HIV testing-related capital expenditures as well as costs of ART over the study time horizon. The parameter estimates for costing are presented in table 1.

### Analysis

The base case analysis compared the mean costs, correct classification of patients, HIV cases diagnosed, HIV cases per 10 000 base case patients, and DALYs of PITC, SU-PITC and POC-NAAT. We adjusted for disability to enable the calculation of DALYs over the 10-year time horizon using disability weights for symptomatic HIV pre-AIDS and HIV/AIDS on ART obtained from the Global Burden of Diseases study (table 1).[21] We stratified the population of patients by AIDS status (ie, pre-AIDS and AIDS) and adjusted their years of life lived with pre-AIDS and AIDS by their associated disability weights and added the years of life lost due to premature mortality by subtracting mean age at death from the life expectancy in Kenya. This stratification method and published disability weights were used because we lacked patient-specific disability weights (obtained from person trade-off studies).

We calculated incremental cost-effectiveness ratios (ICERs) using cost per additional correctly classified patient, cost per additional HIV case identified, cost per HIV case averted and cost per DALY averted. ICERs were calculated by dividing the difference in costs between pairs of comparators by the difference in outcomes for the same comparators. Given three comparators, the aim of the incremental analysis was to identify the most optimal choice by excluding dominated and extended dominated alternatives and identifying the lowest ICER for each comparator by outcome measure. Dominance occurs when one comparator in an incremental cost-effectiveness (ICE) analysis is less costly and more effective than another comparator. Extended dominance occurs when there are more than two comparators in an ICE analysis and one of the comparators is dominated by a linear combination of two other comparators. We also included analyses comparing SU-PITC and POC-NAAT independently, pairwise, to PITC. This analysis informs a potential policy choice in which only one of either SU-PITC or POC-NAAT were available in a given jurisdiction. As a threshold for cost-effectiveness, we used three times Kenya's per capita gross domestic product (2019), which is $5450 (World Bank).[22]

We conducted univariate sensitivity analyses to identify the most influential parameters on the results of the analysis of the primary outcome (ie, DALYs). All parameters varied across plausible ranges using 95% CIs where available and ranges of ±20% for probabilities and ±50% for costs when 95% CIs were unavailable (table 1). These arbitrary value ranges in the absence of CIs are often used in the literature[23 24] and reflect the higher uncertainty observed in cost estimates compared with estimates of other parameters. We used Monte Carlo simulation to conduct probabilistic sensitivity analyses aimed at assessing overall parameter uncertainty in the model and further testing the robustness of results. We used baseline values as means and estimated SEs assuming ranges were equivalent to 95% CIs (four times the SE). We assumed beta distributions for probabilities and disability weights and normal distributions for other parameters. The analysis was performed using TreeAge Pro 2021, and this report conforms to the Consolidated Health Economic Evaluation Reporting Standards statement.[25]

## RESULTS

### Base case analysis

Results of the baseline analysis are shown in table 2. The mean cost of HIV testing per participant (base case patient modelled) under PITC was $0.50, compared with $1.87 for SU-PITC, and $29.83 for POC-NAAT. The mean cost of HIV-positive patient management per participant under PITC was $46.74, compared with $171.72 for SU-PITC, and $185.08 for POC-NAAT. The total cost of PITC (including both testing and HIV-positive patient management) per participant was therefore $47.24, compared with $173.59 for SU-PITC, and $214.91 for POC-NAAT. Therefore, SU-PITC increased average costs per participant by $126.34 relative to PITC and POC-NAAT increased costs by $41.30 relative to SU-PITC.

Of all base case patients, 25.3% were correctly classified under PITC, compared with 93.7% under SU-PITC and 94.9% under POC-NAAT (correct classification was a factor of both the HIV test being performed and the accuracy of the testing strategy). The increase in correct classification comparing SU-PITC to PITC (due to scaling up coverage with the same HIV rapid tests from 25.6% to 94.9%) was 68.4%, and the increase in correct classification comparing POC-NAAT to SU-PITC (due, predominantly, to diagnosis of acute

**Table 2** Mean base case patient costs and outcomes of HIV testing strategies to detect acute and prevalent HIV infection in adult outpatients seeking healthcare for symptoms compatible with acute HIV infection in Kenya

| | PITC | SU-PITC | POC-NAAT |
|---|---|---|---|
| **Costs ($)** | | | |
| Testing costs | 0.50 | 1.87 | 29.83 |
| HIV-positive patient management | 46.74 | 171.72 | 185.08 |
| Total costs | 47.25 | 173.59 | 214.91 |
| Incremental cost | – | 126.34 | 41.32 |
| **Testing** | | | |
| Proportion accurately classified* | 0.2530 | 0.9374 | 0.9488 |
| Incremental accurate classification | – | 0.6844 | 0.0114 |
| Cost per additional accurately classified case | – | $184.59 | $3619 |
| HIV-positive case yield | 0.0125 | 0.0460 | 0.0465 |
| Incremental HIV-positive case yield | – | 0.0335 | 0.0005 |
| Cost per additional HIV-positive case identified | – | $3774 | $83 053 |
| **HIV prevention** | | | |
| HIV cases (per 10 000) | 606 | 596 | 545 |
| HIV cases averted | – | 10 | 51 |
| Cost per HIV case averted | – | Dominated† | $2.75‡ |
| HIV cases averted (reference to PITC) | – | 10 | 61 |
| Cost per HIV case averted (reference to PITC) | – | $12.63 | $2.75 |
| **Main outcome** | | | |
| DALYs | 0.2140 | 0.1757 | 0.1598 |
| DALYs averted | – | 0.0382 | 0.0159 |
| Cost per DALY averted | – | Dominated† | $3,098‡ |
| DALYs averted (reference to PITC) | – | 0.0382 | 0.0541 |
| Cost per DALY averted (reference to PITC) | — | $3306 | $3098 |

*A factor of both the decision to test and diagnostic test accuracy.
†SU-PITC is excluded from the incremental analysis by extended dominance.
‡Incremental cost-effectiveness ratio compares POC-NAAT to PITC.
Ab, antibody; DALY, disability-adjusted llife-year; PITC, provider-initiated testing and counselling; POC-NAAT, point-of-care nucleic acid amplification testing; SU-PITC, scaled-up provider-initiated testing and counselling.

cases) was 1.1%. The ICER comparing SU-PITC to PITC was $185 per additional correctly classified case, and the ICER comparing POC-NAAT to SU-PITC was $3619 per additional correctly classified case.

PITC had an HIV-positive case yield of 1.25%, compared with 4.60% under SU-PITC and 4.65% under POC-NAAT. The increase in HIV-positive case yield comparing SU-PITC to PITC was 3.35%, while the increase in HIV-positive case yield comparing POC-NAAT to SU-PITC was 0.05%. The ICER comparing SU-PITC to PITC was $3774 per additional HIV case identified, while the ICER comparing POC-NAAT to SU-PITC was $83 052 per additional HIV case identified.

In the model, the PITC testing strategy was projected to result in 606 HIV cases per 10 000, compared with 596 HIV cases for the SU-PITC testing strategy and 545 HIV cases for POC-NAAT testing strategy. Therefore, SU-PITC was projected to avert 10 HIV cases compared with PITC, and POC-NAAT was projected to avert 51

cases compared with SU-PITC. In the incremental analysis of cases averted, SU-PITC was eliminated due to extended dominance (ie, the cases averted by SU-PITC are dominated by a linear combination of the cases averted by PITC and POC-NAAT). The ICER comparing POC-NAAT to PITC was $2.75 per HIV case averted (the ICER comparing POC-NAAT to SU-PITC is not calculated in the presence of extended dominance). In an analysis assuming SU-PITC as the only available option, the ICER comparing SU-PITC to PITC is $12.63 per HIV case averted.

With respect to DALYs, the primary outcome of interest, PITC was associated with a mean of 0.214 DALYs over the 10-year time horizon, compared with 0.176 DALYs for SU-PITC and 0.160 DALYs for POC-NAAT. Therefore, SU-PITC averted 0.038 DALYs compared with PITC and POC-NAAT averted 0.016 DALYS compared with SU-PITC. In the incremental analysis, SU-PITC was eliminated by extended dominance given

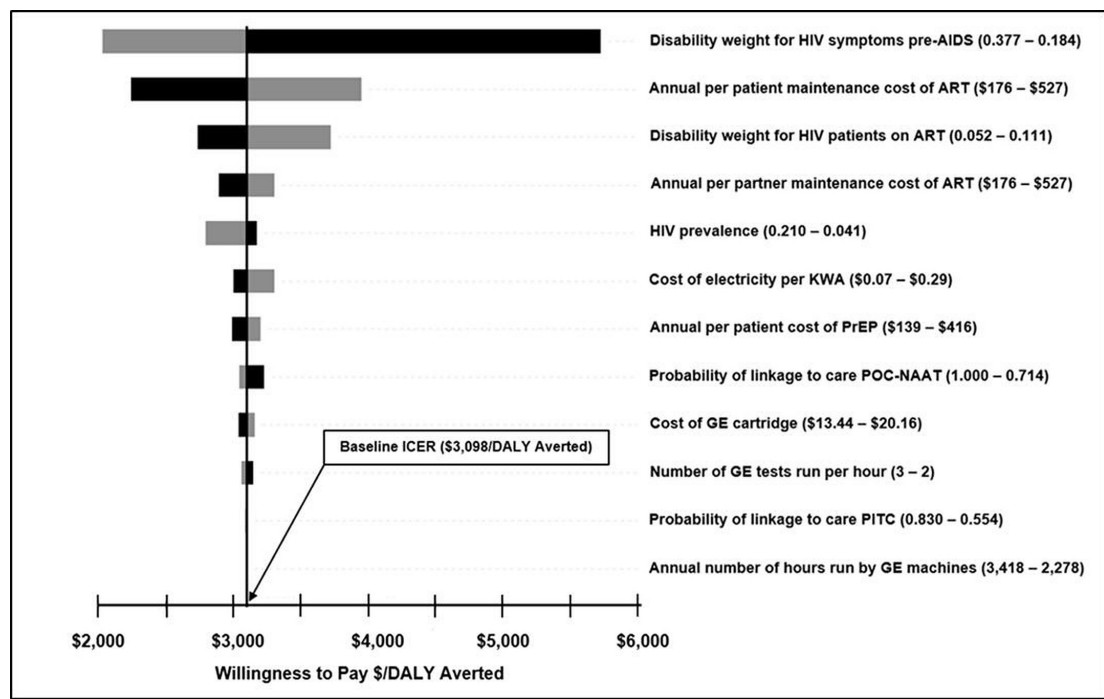

**Figure 4** Tornado diagram of univariate sensitivity analysis showing the model parameters with the most influence on the cost per DALY averted comparing POC-NAAT and PITC. ART, antiretroviral therapy; DALY, disability-adjusted life-year; GE, GeneXpert,KWA, kilowatt hour; PITC, provider-initiated testing and counselling; POC-NAAT, point-of-care nucleic acid amplification testing; PrEP, pre-exposure prophylaxis.

that the DALYs averted by SU-PITC are dominated by a linear combination of the DALYs averted by PITC and POC-NAAT. The ICER comparing POC-NAAT to PITC was $3098 per DALY averted. In a pairwise analysis comparing SU-PITC to PITC, an analysis that assumes that SU-PITC was the only available alternative to PITC, the ICER was $3306 per DALY averted.

## Sensitivity analysis

The results of the one-way sensitivity analysis are shown as Tornado diagrams in figure 4. The ICER comparing POC-NAAT to PITC ($/DALY averted)

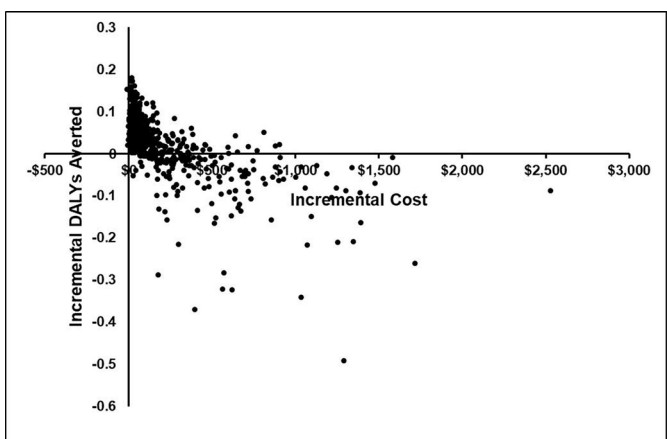

**Figure 5** Incremental cost-effectiveness scatterplot of probabilistic sensitivity analysis showing the distribution of cost-effectiveness pairs on the cost-effectiveness plane. DALYs, disability-adjusted life-years.

was most sensitive to the disability weight for symptomatic HIV pre-AIDS, the annual cost of ART for patients who tested positive for HIV, the disability weight for HIV/AIDS receiving ART, and the annual per patient cost of ART provided to seropositive partners (figure 4).

The results of the probabilistic sensitivity analysis are shown in figure 5 as an incremental cost-effectiveness (ICE) scatterplot and figure 6 as a cost-effectiveness acceptability curve (CEAC) for the comparison of POC-NAAT and PITC. The ICE scatterplot shows that there was substantial uncertainty as to the impact of POC-NAAT on averting DALYs compared with PITC but near zero uncertainty as to the cost increase of POC-NAAT compared with PITC.

The CEAC shows the probability that PITC and POC-NAAT would be cost-effective in 1000 runs of Monte Carlo simulation. PITC has a higher probability of being cost-effective than POC-NAAT up to a willingness to pay of just under $2000 per DALY averted at which point POC-NAAT exhibits a higher probability of being cost-effective. At a willingness to pay of $5450 per DALY averted (the threshold used in the analysis), the distribution of cost-effectiveness in this simulation was 21.9% PITC vs 40.0% POC-NAAT. The probability that PITC would be cost-effective diminishes with increasing willingness to pay, while the probability that POC-NAAT would be cost-effective increases and peaks at 52% at a willingness to pay of approximately $55 000 per DALY averted (figure 6).

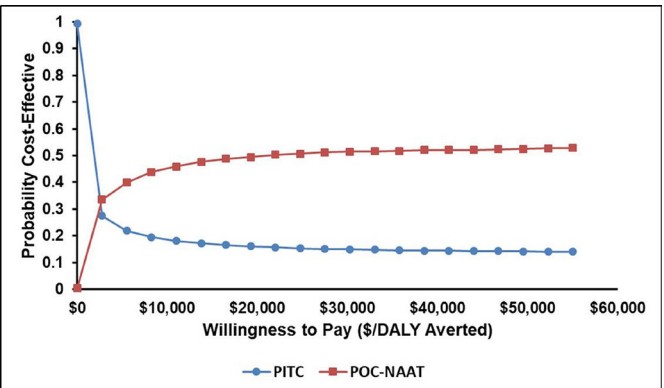

**Figure 6** Cost-effectiveness acceptability curve showing the probability that POC-NAAT is cost-effective compared with PITC at varying levels of willingness to pay for a DALY averted. DALY, disability-adjusted life-year; PITC, provider-initiated testing and counselling; POC-NAAT, point-of-care nucleic acid amplification testing.

## DISCUSSION

We assessed the cost-effectiveness of testing strategies to diagnose only chronic HIV infection (including PITC and scaled-up antibody testing (SU-PITC)) versus a testing strategy to diagnose both acute and chronic HIV infection (ie, POC-NAAT) in symptomatic adults meeting test criteria in outpatient primary care facilities in Kenya. In terms of correct patient classification, a factor of the decision to test as well as the accuracy of the diagnostic strategy, and HIV-positive patient yield, expanding coverage of rapid antibody testing by testing all patients that meet risk criteria unless they opt out (SU-PITC) was more cost-effective than PITC, the current standard of care and a testing strategy including HIV-1 RNA testing that was able to diagnose AHI (POC-NAAT). However, in terms of HIV cases averted, a measure of HIV prevention, and DALYs averted, a metric that includes both length and quality of life, POC-NAAT was more cost-effective than both PITC and SU-PITC.

With an ICER of $3306 per DALY averted, and SU-PITC eliminated from consideration due to extended dominance, POC-NAAT was cost-effective when considering the traditional threshold of three times Gross Domestic Product (GDP) per capita: the GDP per capita of Kenya in 2019 was $1817, suggesting a threshold of $5450. However, this GDP-based threshold for cost-effectiveness is considered by some to be too high, and programmes considered cost-effective are often not implemented in the context of country-specific healthcare spending.[26 27] Some studies suggest thresholds as low as 1% of GDP per capita ($18 in Kenya) in low-income countries when no external funding is available.[28–30] It is also worth noting that HIV has and has had significant amounts of external funding and that higher thresholds are apt in this context.

Previous economic evaluations have demonstrated that testing for AHI is cost-effective in high-risk populations, such as men who have sex with men and injection drug users at sufficiently high HIV prevalence (over 0.4%)

both in the USA and in China.[31] We found no study of the cost-effectiveness of diagnosis for AHI using NAAT in sub-Saharan Africa. However, in countries such as Kenya, where the national HIV prevalence is over 4%, and for patients such as those included the TMP trial (ie, adults aged 18–39 presenting for outpatient care with symptoms compatible with AHI—the base case patient in this study), the risk of HIV acquisition may be similar to that in the key populations in which previous studies demonstrated that testing for AHI was cost-effective.[32–34]

In one-way sensitivity analysis, the ICER comparing POC-NAAT to PITC was sensitive to disability weights for HIV and costs of ART. The ICER comparing POC-NAAT to PITC increases substantially (less cost-effective) at low disability weights for asymptomatic patients, reflecting the disadvantage of treating relatively well people. The ICER comparing POC-NAAT to PITC was also sensitive to variations in per patient costs of ART both for index patients and partners, with higher costs leading to higher ICERs (less cost-effective). This is to be expected because ART is for a lifetime and early diagnosis of HIV and the impact of ART to extend life increase the overall cost of ART. Surprisingly, varying the costs of GeneXpert cartridges often considered high in low-income settings did not affect the ICER comparing POC-NAAT to PITC substantially (figure 4), although low cartridge prices would be desirable from a budget impact (affordability) perspective.

The antibody tests used in the TMP trial, Determine and First Response, have near-perfect sensitivity and specificity for diagnosis of chronic HIV[35 36] but are unable to detect AHI. Cepheid's Xpert HIV-1 Qual platform has high sensitivity and specificity for diagnosis of chronic untreated HIV, with the added benefit of diagnosis of AHI.[37] Despite this ability of NAAT with the Xpert platform to identify AHI, scaled-up PITC by opt-out in the setting of out-patient clinics (SU-PITC) using rapid antibody tests performed well in terms of both accurate classification of patients and yield of HIV-positive patients, as the majority of infections are chronic (see table 2). Therefore, SU-PITC, though not the most cost-effective in terms of cost per DALY averted (the primary outcome), is worth consideration for widespread scale-up a to improve overall accurate classification of patients and yield of HIV positive case relative to the status quo (PITC), despite missing AHI. This point is further reinforced by the results of the probabilistic sensitivity analysis which shows substantial uncertainty in effectiveness (DALYs averted) comparing POC-NAAT and PITC (figure 5). In this analysis, the identification of AHI using POC-NAAT was not as robust in its increase in effectiveness as it was in its increase in cost. If the cost of RNA testing could be reduced (eg, through reduced cost of GeneXpert cartridges), POC-NAAT would become a more consistently attractive option.

This economic evaluation is generalisable to other, similar countries with high prevalence and populations vulnerable to HIV as assessed by risk scoring at outpatient health facilities. This is because the cost of testing,

including the cost of GeneXpert cartridges and antibody tests, is unlikely to vary substantially given that prices are set internationally and countries are exposed to uniform prices, and the costs of other inputs (such as personnel), though variable, were not drivers of cost-effectiveness in the analysis. Most other countries in sub-Saharan Africa that resemble Kenya in this regard face the same issues: how to pay for scaled-up antibody testing or testing to detect AHI given extreme budget constraints. Given the preventive impact on the HIV epidemic, NAAT testing for AHI in primary care settings should be considered strongly despite higher costs, particularly where dedicated HIV funding is available.

This analysis had a number of strengths. This is the first study in sub-Saharan Africa to consider the cost-effectiveness of NAAT testing for HIV including AHI among symptomatic outpatients. We also included three alternative testing strategy to maximise the policy relevance and potential utility of the study results. One limitation of the analysis was that we conducted the analysis over a time horizon of 10 years and not a lifetime horizon. In using a shorter time horizon, we favoured short-term certainty in our modelling over long-term uncertainty, given that patients with HIV now have life expectancies approaching those of the general population. Additionally, while we performed univariate and probabilistic sensitivity analyses to account for parameter uncertainty in the decision tree model, we did not account for the additional uncertainty in the number of future HIV cases averted which were obtained from the stochastic network-based model. Lastly, alternative methods to diagnose AHI (eg, rapid antigen–antibody tests) were not considered in this cost-effectiveness analysis.

As community HIV testing using antibody tests increases in coverage and AHI increases in importance as a driver of HIV incidence, HIV-1 NAAT for AHI at the point of care for patients vulnerable to HIV and meeting AHI screening criteria will increase in importance as a diagnostic mechanism. Our analysis has shown that HIV-1 NAAT for AHI is cost-effective, subject to willingness to pay and the availability of adequate funding. To increase testing coverage and achieve the UNAIDS policy goal of diagnosing 95% of persons living with HIV/AIDS, countries like Kenya will face higher costs because the marginal cost of identifying higher percentages of PLWH increases as testing coverage increases. These higher up-front testing costs may be offset, however, by savings related to HIV cases prevented, especially if prompt linkage to care and effective ART limit HIV transmission from index cases. In conclusion, increasing testing coverage for patients vulnerable to HIV at outpatient clinics is cost-effective and will become increasingly so with increasing healthcare budgets and reducing costs of HIV-1 NAAT platforms.

**Author affiliations**
[1]Saw Swee Hock School of Public Health, National University Singapore, Singapore
[2]KEMRI-Wellcome Trust Research Programme, Kilifi, Kenya
[3]Center for Studies in Demography and Ecology, University of Washington, Seattle, Washington, USA
[4]Department of Medicine, University of Washington, Seattle, Washington, USA
[5]Africa Health Research Institute, Durban, South Africa
[6]National AIDS & STI Control Programme, Nairobi, Kenya
[7]National Institutes of Allergy and Infectious Diseases, National Institutes of Health, Rockville, Maryland, USA
[8]Departments of Anthropology and Epidemiology, University of Washington, Seattle, Washington, USA
[9]Nuffield Department of Medicine, University of Oxford, Headington, UK
[10]Departments of Medicine, Global Health, and Epidemiology, University of Washington School of Medicine, Seattle, Washington, USA

**Contributors** JBB: study conception and design, design of decision-analytic model, analysis of primary data, development and parametrisation of the decision-analytical model, preparation of the first draft of the manuscript, review and revision of drafts of the manuscript, and guarantor of the study. CAA: leadership, collection and analysis of primary data; and review and revision of drafts of the manuscript. DTH: development of stochastic agent-based mathematical model for parametrisation of the decision-analytical model; parametrisation of decision model; and review and revision of drafts of the manuscript. EvdE: primary data collection; and review and revision of drafts of the manuscript. AH: primary data collection and review and revision of drafts of the manuscript. EG, PMM, TN, MS and WC: primary data collection; and review and revision of drafts of the manuscript. CF: study conception and design, and review and revision of drafts of the manuscript. SMG: development of stochastic agent-based mathematical model for parametrisation of the decision-analytical model and review and revision of drafts of the manuscript. EJS: study conception and design; conduct of trial (Tambua Mapema Plus (TMP)) which formed the main parametrisation basis for the decision-analytical model; parametrisation of the decision-analytical model; and review and revision of drafts of the manuscript. SMG: study conception and design, design of decision-analytical model; parametrisation of decision-analytical model, conduct of trial (TMP) which formed the main parametrisation basis for the decision-analytical model; and review and revision of drafts of the manuscript. All authors: review, revision and approval of the final draft of the manuscript.

**Funding** Funding for the Tambua Mapema Trial was provided by the National Institute of Allergy and Infectious Diseases (NIAID), grant R01 AI124968. SMG was also supported by the Robert W. Anderson Endowed Chair in Medicine. We acknowledge Usha Sharma of the NIAID Division of AIDS for the scientific support for the Tambua Mapema Plus study. This work was also supported through the Sub-Saharan African Network for TB/HIV Research Excellence, a Developing Excellence in Leadership, Training and Science (DELTAS) Africa Initiative (grant no. DEL-15–006). The DELTAS Africa Initiative is an independent funding scheme of the African Academy of Sciences's Alliance for Accelerating Excellence in Science in Africa and supported by the New Partnership for Africa's Development Planning and Coordinating Agency with funding from the Wellcome Trust (grant number 107752/Z/15/Z) and the UK government. Truvada was supplied by Gilead Sciences.

**Competing interests** None declared.

**Patient and public involvement** Patients and/or the public were not involved in the design, conduct, reporting or dissemination plans of this research.

**Patient consent for publication** Not applicable.

**Ethics approval** Not applicable.

**Provenance and peer review** Not commissioned; externally peer reviewed.

**Data availability statement** Data are available upon reasonable request.

## ORCID iDs

Joseph B Babigumira http://orcid.org/0000-0003-3834-7141
Peter Mwangi Mugo http://orcid.org/0000-0002-1808-3292
Susan M Graham http://orcid.org/0000-0001-7847-8686

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
