## [Reviewer comments · BMJ Open]

ARTICLE DETAILS

TITLE (PROVISIONAL)	Testing strategies to detect acute and prevalent HIV infection in adult outpatients seeking healthcare for symptoms compatible with acute HIV infection in Kenya: a cost-effectiveness analysis
AUTHORS	Babigumira, Joseph; Agutu, Clara A.; Hamilton, Deven T.; van der Elst, Elise; Hassan, Amin; Gichuru, Evanson; Mugo, Peter; Farquhar, Carey; Ndung'u, Thumbi; Sirengo, Martin; Chege, Wairimu; Goodreau, Steven M; Sanders, Eduard; Graham, Susan

VERSION 1 – REVIEW

REVIEWER	Hatzold, Karin Population Services International, HIV and TB
REVIEW RETURNED	20-Dec-2021

GENERAL COMMENTS	Excellent, highly valuable manuscript.
--

REVIEWER	Fenny, Ama University of Ghana
REVIEW RETURNED	15-Mar-2022

GENERAL COMMENTS	Methods Outcomes  1. There is no mention of cost in the statement of study outcomes. Thus, it is not clear whether, for example, outcome 4 is a cost per DALY averted or not. 2. It is mentioned in the outcome that there was an adjustment for disability to enable calculation of DALY over a 10-year horizon. The statement is inappropriately overused, and in this context, misplaced and must be shifted to the statistical analysis section. It must explain how the adjustment was done. Was it by stratification, standardization, etc., and why? Decision model  3. I struggle to read the figure depicting the decision-analytic model because the text is illegible. Kindly resolve that. Some items on the CHEERS checklist are unconditional and cannot be omitted or referred to as Not Applicable. Therefore, their omission makes the methods section lack clarity and needs resolving. Design
---

4. All economic evaluation studies are either single study-based or synthesis-based. I see the former being the case and should be indicated and explained as required in item 11a of the CHEERS checklist. (Example, the paper mention in the design that the study is a model-based cost effectiveness analysis using cost and outcomes data obtained from the TMP trial, which supposedly was conducted by the authors but again indicated on the CHEERS checklist that neither item 11a or 11b was applicable to the study. That is very confusing.

Measurement of effectiveness

5. The statistical analysis should come after statements of the measurement of effectiveness, which is unconditional in cost-effectiveness analysis. For example, we measured the effectiveness of outcomes by doing A, B, C, and D. Or take each outcome at a time and explain how you measured effectiveness. (Same must be included in the methods section of the abstract)

Statistical analysis

6. This section needs restructuring and must follow the stated procedure for measuring the effectiveness of the outcomes. I suggest taking each study outcome and describing the statistical method used with a justification. That will make it clearer for everyone to comprehend in a step-by-step manner and also for replication. Example, if you say “the base case analysis compared the mean cost, correct classification of patients, HIV cases diagnosed, HIV cases per 10,000 base-case patients, and DALY of the PITC, etc.” For each instance, how was the mean estimated? Which statistical procedure did you use? Was the mean estimate accompanied by either SD or 95%CI? It has become increasingly important to provide a vivid explanation in statistical methods to aid the selection of papers for meta-analyses and further studies.
7. Also, clearly state how you analyzed the ICER. Do not assume everyone knows how to compute ICER because your audience may not have technical knowledge of Health economics or health economic evaluation.

Sensitivity analysis

8. Justify the arbitrary use of the +/- 20% and 50% range for mean parameter values in situations where they were not available. That is, what informed the decision for the choice of uncertainty interval in such circumstances where they were not available?
9. The input parameters are a lot, but there is no justification for reporting a few in the one-way sensitivity graph (Tornado diagram). Did the other

	variables not showing in the graph not sensitive to the base-case cost? The best suggestion will be to use the end-point cost per patient for each cohort in the sensitivity analysis rather than the disaggregated cost, which is insignificant and feed into the total cost. Currency conversion 10. There is a new normal for currency conversion in economic evaluation. Using CPI for cost conversion defeats the purpose for international comparison and needs resolving by using a standardized purchasing power parity converter, which is accessible here https://eppi.ioe.ac.uk/costconversion/ Other comments on the methods  ➤ The study refers to an unpublished trial paper for additional information on the method, which makes it difficult for anyone to access. Even if the trial paper is now published, which is not, the authors must state briefly what the mathematical modeling quoted in the outcome section and other aspects of the methods were about and why. ➤ There is an overlap of study focus, making it difficult to understand what exactly was measured. Perhaps, it will be better to simply compare outcomes for standard care (PITC) with the new technology, that is point-of-care nucleic acid amplification testing (POC-NAAT). At one point it is expanded PITC, and expanded POC-NAAT. Consistent use of terminology is required. ➤ Since the base-case analysis was based on the trial and evaluated concurrently for standard care and the intervention, Table 2 should present the mean and 95%CI for the end-point effect and costs with an additional column for differences in outcomes as a measure of the incremental outcomes. That makes it much clearer to understand than it is now. ➤ Because I attempted to access the trial paper, I found that the referenced number 35 is missing a lot of information because a published protocol for the study has many authors (doi: 10.2196/16198) but missing in the referenced trial paper.
--	---

VERSION 1 – AUTHOR RESPONSE

Below is a point-by-point response to the issues, with reviewer comments in italics.

1. **Methods, Outcomes.** *There is no mention of cost in the statement of study outcomes. Thus, it is not clear whether, for example, outcome 4 is a cost per DALY averted or not.*

Thank you for this feedback. We have revised the methods section in response to the feedback received, and now explicitly call out cost as an outcome under the subheading "Outcomes" on line 208, page 6 (Track Changes version).

2. **Methods, Outcomes.** *It is mentioned in the outcome that there was an adjustment for disability to enable calculation of DALY over a 10-year horizon. The statement is inappropriately overused, and in this context, misplaced and must be shifted to the statistical analysis section. It must explain how the adjustment was done. Was it by stratification, standardization, etc., and why?*

In response to this concern, we have moved the information on disability adjustment to the "Analysis" section and specified our approach to this adjustment and the rationale for the method used.

3. **Methods, Decision model.** *I struggle to read the figure depicting the decision-analytic model because the text is illegible. Kindly resolve that.*

We have separated the components of Figure 1 into 3 separate files, decreased the white space and increased the font size, so that it is easier to read. We have uploaded these files and updated the captions.

4. **Methods, Design.** *Some items on the CHEERS checklist are unconditional and cannot be omitted or referred to as Not Applicable. Therefore, their omission makes the methods section lack clarity and needs resolving.*

Thank you for this feedback. We have carefully addressed the points related to the analyses performed and ensured that we address all items on the CHEERS checklist. Please see attached an amended CHEERS checklist with track changes to reflect our updates.

All economic evaluation studies are either single study-based or synthesis-based. I see the former being the case and should be indicated and explained as required in item 11a of the CHEERS checklist. (Example, the paper mention in the design that the study is a model- based cost effectiveness analysis using cost and outcomes data obtained from the TMP trial, which supposedly was conducted by the authors but again indicated on the CHEERS checklist that neither item 11a or 11b was applicable to the study. That is very confusing.

We agree that all economic evaluation studies are either single-study based or synthesis-based. The study was not a single study-based economic evaluation. We did leverage the Tambua Mapema clinical trial as a source of parameters to inform the model-based analysis, but it was not the only source of parameters. In addition to data from the trial, we used estimates from a mathematical model as well as the published literature as described under “Study Design”. In response to this concern, we have updated the CHEERS checklist accordingly, specifying that the analysis was, indeed, a model-based analysis. We have also made changes to the Methods (Study design) (lines 129 – 138 (Track Changes version)) to emphasize that this study was a synthesis- and model-based analysis.

5. Methods, Measurement

of effectiveness. *The statistical analysis should come after statements of the measurement of effectiveness, which is unconditional in cost-effectiveness analysis. For example, we measured the effectiveness of outcomes by doing A, B, C, and D. Or take each outcome at a time and explain how you measured effectiveness. (Same must be included in the methods section of the abstract)*

The different outcomes (costs, DALYs, diagnostic accuracy, case yield, cases averted) were calculated by rolling back the decision-analytic model. As such, we did not conduct frequentist analyses as would be required if we performed a statistical analysis of a single study-based & trial-based cost-effectiveness analysis. Our analyses of primary data as described throughout our Methods section involve the generation of parameter estimates from primary data and the use of estimates from published sources to parameterize the decision model. As such, our effectiveness measures come from the model as described under “Decision analytic model” (lines 175 – 204).

- 6. Methods, Statistical analysis.** *This section needs restructuring and must follow the stated procedure for measuring the effectiveness of the outcomes. I suggest taking each study outcome and describing the statistical method used with a justification. That will make it clearer for everyone to comprehend in a step-by-step manner and also for replication. Example, if you say “the base case analysis compared the mean cost, correct classification of patients, HIV cases diagnosed, HIV cases per 10,000 base-case patients, and DALY of the PITC, etc.” For each instance, how was the mean estimated? Which statistical procedure did you use? Was the mean estimate accompanied by either SD or 95%CI? It has become increasingly important to provide a vivid explanation in statistical methods to aid the selection of papers for meta-analyses and further studies.*

To address issues relating to statistical analysis, we emphasize that the study was a model-based analysis in both the “Abstract” and under “Study Design”, with effectiveness measures generated by analyzing (rolling back) the decision-analytic model. We did not conduct outcome-specific frequentist analyses using primary data from the trial. The data were synthesized by the decision-analytic model through input parameters. Therefore, this study does not qualify to be included in a synthesis of data on outcomes as would be performed in a meta-analysis. When we refer to 10,000 base case patients, this number is included as a parameter in the decision modeling and the model produces outcomes for the number of patients included.

7. **Methods, Statistical analysis.** *Also, clearly state how you analyzed the ICER. Do not assume everyone knows how to compute ICER because your audience may not have technical knowledge of Health economics or health economic evaluation.*

In response to this feedback, we have added a statement of how we calculated the ICER, explaining the configuration of the ICER as the difference in costs divided by the difference in outcomes between pairs of comparators (lines 277 – 278 (Track Change version)).

8. **Methods, Sensitivity analysis.** *Justify the arbitrary use of the +/- 20% and 50% range for mean parameter values in situations where they were not available. That is, what informed the decision for the choice of uncertainty interval in such circumstances where they were not available?*

We have added a justification for the choice of uncertainty interval for model parameters for which there were no reported 95% confidence intervals in the sources of our estimates. These arbitrary ranges are **often used in the literature** and reflect the tendency of cost estimates to be subject to significantly more uncertainty than other estimates such as probabilities. We believe that these ranges are wide enough to account for parameter uncertainty in univariate and probabilistic sensitivity analyses. We added two citations^{1 2} as examples of published studies that use these parameter ranges.

9. **Methods, Sensitivity analysis.** *The input parameters are a lot, but there is no justification for reporting a few in the one-way sensitivity graph (Tornado diagram). Did the other variables not showing in the graph not sensitive to the base-case cost? The best suggestion will be to use the end-point cost per patient for each cohort in the sensitivity analysis rather than the disaggregated cost, which is insignificant and feed into the total cost.*

The variables included in the Tornado diagram of one-way sensitivity analysis represent, as suggested by the reviewer, the most impactful variables on the cost per DALY averted, the main outcome of the analysis. These variables are the only ones that impacted the baseline estimate and every other variable was not sensitive to the base case cost per DALY averted and hence

did not appear on the Tornado diagram, The last three variables included are barely impactful and every other variable is not impactful at all.

10. **Methods, Currency conversion.** *There is a new normal for currency conversion in economic evaluation. Using CPI for cost conversion defeats the purpose for international comparison and needs resolving by using a standardized purchasing power parity converter, which is accessible here <https://eppi.ioe.ac.uk/costconversion/>*

We acknowledge the importance of international cost comparison across studies, particularly studies of novel therapeutics to be used in multiple countries and modeling studies in which estimates are obtained from multiple countries. However, the purpose of this study, as written in the Introduction (page 4 lines 22 – 24), was to “support the widespread use of NAAT testing for AHI in the outpatient setting in Kenya.” As such, we thought it necessary to: (1) use estimates from Kenya for all input parameters related to policy, including inputs obtained from the published literature (Table 1), and (2) use Kenya’s CPI for healthcare to reflect local inflationary pressure. This makes the results of our study more suited to and more relevant for local policy decision making.

Other comments on the methods

11. *The study refers to an unpublished trial paper for additional information on the method, which makes it difficult for anyone to access. Even if the trial paper is now published, which is not, the authors must state briefly what the mathematical modeling quoted in the outcome section and other aspects of the methods were about and why.*

This trial and the modeling paper have now been published, and we have updated the references accordingly.

12. *There is an overlap of study focus, making it difficult to understand what exactly was measured. Perhaps, it will be better to simply compare outcomes for standard care (PIT C) with the new technology, that is point-of-care nucleic acid amplification testing (POC-NAAT). At one point it is expanded PITC, and expanded POC-NAAT. Consistent use of terminology is required.*

We apologize for this confusion, as we should not have used the term “expanded POC-NAAT.” To avoid confusion about the three scenarios modeled, we have adopted the same terminology used in our published modeling paper, so that readers who are interested can refer to that publication to learn more about the comparisons made in the model.

13. *Since the base-case analysis was based on the trial and evaluated concurrently for standard care and the intervention, Table 2 should present the mean and 95%CI for the end-point effect and costs with an additional column for differences in outcomes as a measure of the incremental outcomes. That makes it much clearer to understand than it is now.*

The base case analysis was not based exclusively on the trial but on the results of rolling back the decision-analytic model. We therefore do not have results of a frequentist analysis that includes 95% confidence intervals of the base case estimates. Table 2 includes differences in costs and outcomes for each pairwise comparison of comparators.

14. Because I attempted to access the trial paper, I found that the referenced number 35 is missing a lot of information because a published protocol for the study has many authors (doi: [10.2196/16198](https://doi.org/10.2196/16198)) but missing in the referenced trial paper.

As mentioned above, we have replaced this placeholder reference with the actual reference of the trial publication, which has now been published.

VERSION 2 – REVIEW

REVIEWER	Fenny, Ama University of Ghana
REVIEW RETURNED	11-Aug-2022
GENERAL COMMENTS	Comments and have been addressed appropriately.